# Longitudinal changes in glycemic control and associated factors in patients with type 2 diabetes mellitus in a public referral hospital in Peru

Romina Belen Carpio-Martinez[1], Jose Manuel Quiroz-Eugenio[1], Frank Espinoza-Morales[2], Wendy M. Yánac-Tellería[3], Marlon Yovera-Aldana[3]*

**1** Carrera de Medicina Humana, Universidad Científica del Sur, Lima, Perú, **2** CAVIMEDIC Centro de Tecnologías Aplicadas A La Diabetes, Lima, Perú, **3** Grupo de investigación de Neurociencias, Metabolismo, Efectividad Clínica y Sanitaria, Universidad Científica del Sur, Lima, Perú

* myovera@cientifica.edu.pe

## Abstract

### Objectives

To evaluate longitudinal changes in glycemic control, measured by glycated hemoglobin (HbA1c), among patients with type 2 diabetes mellitus treated at a national hospital in Peru, and to identify factors associated with persistent poor glycemic control (HbA1c ≥ 7%).

### Methods

We conducted a retrospective cohort study including 741 patients aged 18–65 years with T2DM who had at least two HbA1c measurements (baseline and one-year follow-up) recorded between 2015 and 2019. Changes in the proportion of patients with HbA1c ≥ 7% were assessed using McNemar's test, and individual baseline-to-follow-up HbA1c changes were evaluated. Factors associated with persistent poor glycemic control over time were examined using generalized estimating equation (GEE) models with binomial distribution and logit link. An exchangeable correlation structure and robust standard errors were applied. Progressive models were constructed according to the percentage of missing data, along with a complete-case sensitivity analysis.

### Results

The proportion of patients with HbA1c ≥ 7% decreased from 66.6% to 60.7% (absolute difference: 5.9%; 95% CI: 2.4–9.4; p < 0.001). Overall, 14.3% of patients improved (from ≥ 7% to < 7%), whereas 8.4% worsened (from < 7% to ≥ 7%), yielding a paired odds ratio of 1.71 (95% CI: 1.23–2.37). The median individual baseline-to-follow-up change in HbA1c was 0.0%; however, 35.6% experienced a reduction

**Data availability statement:** All relevant data are within the manuscript and its Supporting Information files.

**Funding:** Financial support for this study was partially provided by Universidad Científica del Sur (UCSUR) through the Beca Cabieses – Proyectos de Tesis de Pregrado program (grant number PRE-15-2024-00188), awarded to the authors RBCM and JMQE. The official website of the funder is https://www.cientifica.edu.pe. The funder had no role in the study design, data collection and analysis, interpretation of results, decision to publish, or preparation of the manuscript. If the manuscript is accepted for publication, the university may provide support to cover the article processing charge (APC); this potential support will not influence any aspect of the editorial process.

**Competing interests:** The authors have declared that no competing interests exist.

>0.5%, and 28.3% an increase >0.5%. In GEE models, the odds of HbA1c ≥7% were lower at follow-up (adjusted OR: 0.73; 95% CI: 0.61–0.88). Longer diabetes duration (≥10 years), insulin use, and hypertriglyceridemia were associated with higher odds of poor glycemic control, whereas reduced estimated glomerular filtration rate was associated with lower odds.

## Conclusions

Among patients with type 2 diabetes treated at a public hospital in Peru, a modest reduction in the proportion of poor glycemic control was observed after one year of follow-up. Although this reduction was statistically significant, most patients remained above recommended HbA1c targets. These findings underscore persistent gaps in diabetes management and highlight the need for strengthened clinical and public health strategies to improve glycemic control in routine care settings in Peru.

## Introduction

Despite advances in medicine, millions of people diagnosed with type 2 diabetes mellitus (DM2) worldwide remain far from achieving optimal control of their condition [1]. Only modest improvements in glycemic control have been achieved worldwide, with an average annual reduction of just 0.5% in HbA1c levels [2] Likewise, even in high-income countries, glycemic control remains suboptimal: in regions such as the United States and Europe, fewer than 40% of patients achieve an HbA1c level <7%, and this proportion is even lower in emerging countries, where only about one third reach this target [3]. In South America, fewer than half of patients achieve adequate glycemic control [4]. In Peru, the situation is no different [5]. Despite the availability of international and national guidelines that prioritize HbA1c monitoring as a key indicator, clinical outcomes remain unsatisfactory.

Several factors contribute to the gap between routine clinical practice and recommended glycemic targets. The most relevant include longer disease duration, poor adherence to pharmacological treatment, low educational attainment, and limited disease-related knowledge [6]. In addition, comorbid conditions such as obesity, arterial hypertension, and dyslipidemia further complicate comprehensive diabetes management; and complexity of insulin regimens may also promote clinical inertia and increase variability in clinical outcomes [7]. However, in the Peruvian context, evidence on the effectiveness of standard treatments in achieving glycemic control targets among patients treated in public hospitals remains scarce, particularly given that the Ministry of Health defines an HbA1c target of <7%, with follow-up assessments every 3–6 months, as the national therapeutic goal [8].

In light of this knowledge gap, there is a need for a more precise characterization of longitudinal changes in HbA1c and for the identification of clinical and therapeutic factors associated with successful or unsuccessful metabolic control. Accordingly, this study aimed to assess longitudinal changes in glycated hemoglobin among patients with type 2 diabetes mellitus treated at the endocrinology service of a public hospital

in Peru over a one-year follow-up period, and to identify factors associated with the persistence of poor glycemic control (HbA1c ≥ 7%). These findings provide relevant evidence on the effectiveness of current management strategies and may inform the optimization of clinical interventions and resource allocation to improve glycemic control and prevent long-term complications.

## Methods

### Study design and clinical setting

This study was reported in accordance with the STROBE (Strengthening the Reporting of Observational Studies in Epidemiology) guidelines for observational studies. We conducted a retrospective longitudinal observational cohort study using the database of the endocrinology outpatient clinic at Hospital María Auxiliadora administered by the Ministry of Health, The hospital provides medical care to a population originating from surrounding districts in the south of Lima,

### Study population and sample

The sampling frame consisted of adult patients (18–65 years) with a diagnosis of type 2 diabetes mellitus who were treated in the outpatient setting between 2015 and 2019 and were registered in the Diabetes Epidemiological Surveillance Program of the General Directorate of Epidemiology. Patients with only a single recorded evaluation and those lacking a glycated hemoglobin measurement at one year of follow-up were excluded.

### Sampling and sample power

A census sampling approach was used, including all accessible individuals recorded in the database.

The statistical power of the final sample was calculated based on the proportion of discordant pairs, using as reference the study by López Huamanrayme and the power paired proportions command in Stata 19.5 (College Station, Texas, USA). In that study, 10.2% of patients showed improvement (transition from HbA1c ≥ 7% to <7%), while 5.1% worsened (from HbA1c < 7% to ≥7%) [9]. Using these parameters (p21 = 0.102, p12 = 0.051), a 5% significance level, and a sample size of 741 subjects, the estimated power of McNemar's test was 94.6% (power = 0.9455), exceeding the 80% threshold. Additionally, power simulations were performed by varying the number of participants (500–1000) and the proportions of discordant pairs, showing that power remained high and tended to stabilize at values close to 1 as sample size and the difference between the proportions of improvement and worsening increased (S1 Fig).

### Variables and operationalization

The primary outcome was the change in glycemic control at one year, defined as the change in the proportion of patients with HbA1c ≥ 7% between baseline and final assessments. Secondary outcomes included: (i) paired transitions between glycemic control categories (improvement: ≥ 7% to <7%; worsening: <7% to ≥7%; maintained <7%; maintained ≥7%) and the corresponding McNemar odds ratio among discordant pairs; and (ii) the individual change in HbA1c, expressed as ΔHbA1c = baseline − final (positive values indicating a reduction in HbA1c). This Δ was described as a continuous variable (median [IQR]) and categorized into three clinically important groups changes: decrease (<−0.5%), minimal change (−0.5% to 0.5%), and increase (>0.5%).

Sociodemographic characteristics were measured at baseline: age (≥60 vs < 60 years), sex (male/female), educational level (primary or less; secondary or higher), duration of diabetes (≥10 vs < 10 years), arterial hypertension (yes/no), obesity defined by body mass index (BMI ≥ 30 kg/m2 vs < 30 kg/m2), and abdominal obesity (waist circumference ≥102 cm in men or ≥88 cm in women).

Pharmacological exposure at baseline was operationalized in two ways: (a) by individual drug (metformin, sulfonylureas, insulin, and others), and (b) by initial treatment regimen in four categories: no pharmacological treatment, oral antidiabetic drugs (OADs) only, insulin only, and insulin plus OADs.

Laboratory variables were defined at baseline, using clinical cutoffs: fasting glucose ≥130 mg/dL, 2-hour postprandial glucose ≥180 mg/dL, LDL cholesterol ≥100 mg/dL, low HDL cholesterol (men < 40 mg/dL/ women <50 mg/dL), triglycerides ≥150 mg/dL, creatinine ≥1 mg/dL, albuminuria ≥30 mg/24 h, and creatinine clearance (or eGFR) <60 mL/min.

## Procedures and data collection

Data was obtained from the Diabetes Epidemiological Surveillance Program for the period 2015–2019. Data were evaluated and analyzed from october 1 to december 30, 2024., after approval from the hospital's ethics committee.

The information had been originally collected by physicians from the endocrinology service using a standardized form that included demographic, clinical, and laboratory data, recorded during routine outpatient visits or whenever laboratory tests were requested. Data was registered into the official web-based platform of the program (https://app8.dge.gob.pe/diabetes/).

Database cleaning was performed by identifying and correcting extreme values, outliers, inconsistent entries, and missing cells using Microsoft Excel. Inclusion and exclusion criteria were then applied. Finally, a flow diagram was developed showing the number of patients included in and excluded from the analysis.

## Data processing and statistical analysis

First, baseline characteristics of the sampling frame and the included sample were described using frequencies and percentages. Both groups were compared using the χ2 test or Fisher's exact test, as appropriate, and the percentage of missing data for each variable was also reported.

Statistical analyses were conducted according to the paired study design. To assess the overall change in glycemic control at one year (HbA1c ≥ 7%), baseline and final proportions were compared using McNemar's test, and the absolute difference in proportions was estimated with 95% CI adjusted for pairing.

In addition, transitions between glycemic control categories (improved, worsened, maintained <7%, and maintained ≥7%) were quantified, and the paired McNemar odds ratio was calculated as the ratio of discordant pairs (improved/worsened), with corresponding 95% CI. Individual change in HbA1c was defined as the difference between baseline and final values (Δ = baseline − final; positive values indicate a reduction) and was described as a continuous variable using the median and interquartile range. Paired median comparisons were performed using the Wilcoxon signed-rank test, and quantile regression with robust variances was used to estimate the median Δ and its confidence interval. Additionally, Δ was categorized into three clinical bands (<−0.5%, −0.5% to 0.5%, and >0.5%) and related to categorical changes in glycemic control. Complementary descriptive analyses, such as the distribution of Δ HbA1c and category transition matrices, are presented in the supplementary material.

To identify factors associated with poor glycemic control (HbA1c ≥ 7%) over time, generalized estimating equation (GEE) models with a binomial distribution, logit link, exchangeable working correlation structure, and robust standard errors were applied, accounting for clustering by patient (two observations per subject: baseline and final) [10,11]. Crude and adjusted odds ratios with 95% confidence intervals were reported. The modeling strategy was conducted in stages: Model 1 included only variables with no or minimal missing data; Model 2 incorporated variables with approximately 10% of missing data (such as educational level and body mass index); and Model 3 added variables with up to 35% of missing data (such as triglycerides and estimated glomerular filtration rate). Variables with more than 35% of missing information, including abdominal obesity and albuminuria, were excluded from multivariable models. In the main model, the aim was to retain as many clinically relevant covariates as possible; therefore, in Models 2 and 3, missing values were coded as a specific category ("missing").

Additionally, a sensitivity analysis using complete-case analysis was performed by replicating the same three models but restricting each to observations with complete data for the included variables, without creating "missing" categories, to assess the robustness of the observed associations in the presence of missing data. To reduce collinearity among

medications, pharmacological exposure was primarily modeled as the initial treatment regimen in four mutually exclusive categories (none, oral antidiabetic drugs only, insulin only, and insulin plus oral antidiabetic drugs). As a complementary analysis, an analysis restricted to discordant pairs, such as patients who changed HbA1c category (≥7% to <7% or vice versa) between baseline and final assessments, was conducted using McNemar's test stratified by demographic, clinical, therapeutic, and laboratory subgroups, reporting odds ratios (improved/worsened) with 95% confidence intervals for each stratum.

In analyses of the sampling frame by categories of Δ presented in the supplementary material, comparisons were performed using $\chi^2$ or Fisher's exact tests, and for comparisons of continuous Δ, effect size was also reported using Cohen's d with 95% CI.

All tests were two-sided with a significance level of 0.05. Data processing and statistical analyses were performed using Stata version 19.5 (StataCorp, College Station, Texas, USA).

### Ethical considerations

The project (PRE-15-2024-00188) was reviewed by the Institutional Review Board/Ethics Committee of Universidad Científica del Sur and approved under Certificate No. 526-CIEI-CIENTIFICA-2024. It was also reviewed and approved by the Institutional Ethics Committee of Hospital María Auxiliadora under the Unique Registration Code HMA/CIEI/047/2024.

Access was granted to an anonymized database that did not include names, surnames, medical record numbers, or any other personally identifiable information. The data were stored on a personal computer and were used exclusively for research purposes.

## Results

### Sample selection

Of a total of 6,577 individuals with diabetes mellitus who attended the clinic for the first time between 2015 and 2019, 2,333 had a one-year follow-up evaluation. Subsequently, 1,592 additional patients were excluded due to the absence of a glycated hemoglobin measurement at one year. Finally, 741 patients met the established criteria and were included in the analysis (Fig 1).

Patients with diabetes mellitus were identified at their first consultation between 2015 and 2019. Exclusions were due to lack of follow-up or missing HbA1c at one year. The final sample consisted of patients with complete baseline and follow-up HbA1c data

Compared with the sampling frame, study sample had a higher proportion of adults aged ≥60 years (60.6% vs. 51.5%; p = 0.001), a lower proportion of males (30.4% vs. 35.7%; p = 0.005), a higher educational level (p = 0.001) and dyslipidemia (0.001), greater use of metformin (70.9% vs. 66.0%; p = 0.008), and differences in the number of medications at baseline (p = 0.004 for none). No significant differences were identified in other baseline characteristics (Table 1).

### Baseline characteristics

Among the 741 patients included in the analysis, most were women (69.6%) and aged ≥60 years (60.6%). More than half had a low educational level (52.1%), and nearly one third had diabetes duration longer than 10 years (28.6%). At baseline, approximately two thirds of patients had poor glycemic control (HbA1c ≥ 7%). Lipid abnormalities were highly prevalent, with elevated LDL cholesterol observed in 71.3% and low HDL cholesterol in 59.8% of participants. Regarding treatment, metformin was the most commonly prescribed medication (70.9%), and 28.1% of patients were receiving any insulin therapy. The use of novel antidiabetic agents was minimal. Overall, treatment regimens were predominantly based on oral antidiabetic drugs (Table 1).

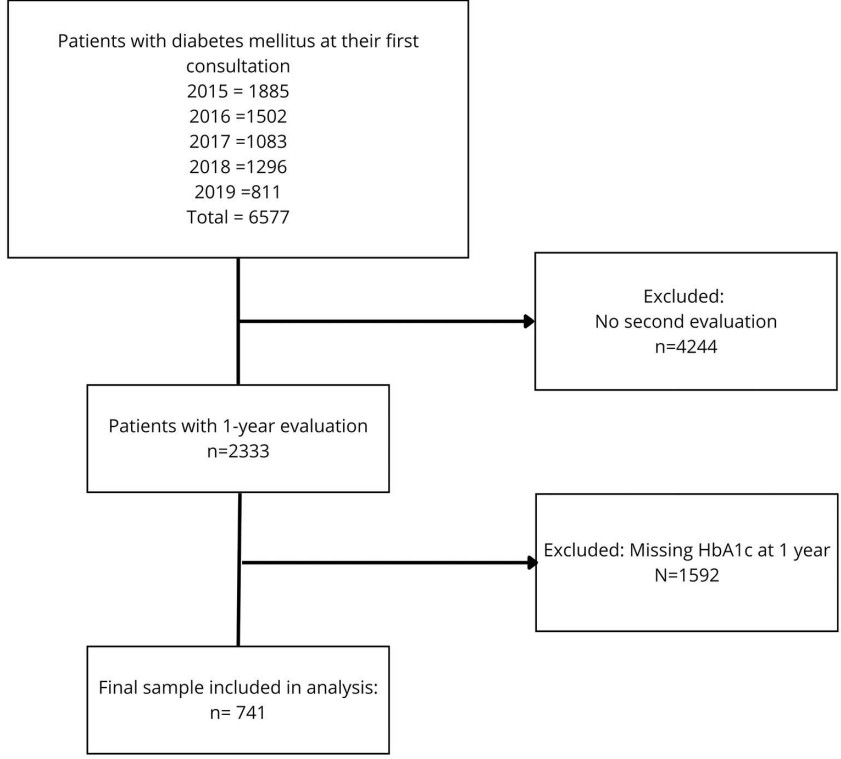

**Fig 1. Flowchart of patient selection for the longitudinal analysis.**

## Overall changes in glycemic control

A significant reduction in the proportion of patients with poor glycemic control (HbA1c ≥ 7%) was observed between the baseline and follow-up assessments (66.6% vs. 60.7%), corresponding to an absolute difference of 5.9% (95% CI: 2.4–9.4). Paired analyses showed that most patients remained with HbA1c ≥ 7% (52.4%), while 25.0% consistently had HbA1c < 7%. Improvement to adequate control (HbA1c ≥ 7% to < 7%) was observed in 14.3% of patients, whereas 8.4% worsened from < 7% to ≥ 7%, yielding a paired odds ratio of 1.71 (95% CI: 1.23–2.37; p < 0.001). Regarding absolute changes in HbA1c values between baseline and follow-up, 43.4% of patients experienced a decrease, 35.9% an increase, and 20.6% showed no change. The median difference was not statistically significant (0; 95% CI: −0.03 to 0.03; p = 1.000). When changes were categorized according to clinical relevance, 35.6% of patients showed improvement (reduction >0.5%), 36.0% exhibited minimal change (−0.5% to 0.5%), and 28.3% experienced worsening (increase >0.5%) (Table 2).

When examining the relationship between categorical changes in glycemic control and the magnitude of individual HbA1c variation, most patients who consistently maintained HbA1c < 7% experienced only minimal changes (−0.5 to 0.5) (65.4%). Among those who remained with HbA1c ≥ 7%, changes were more evenly distributed, with 31.7% showing a clinically meaningful reduction, 36.1% exhibiting minimal variation, and 32.2% experiencing a clinically important increase. In the group that worsened (from HbA1c < 7% to ≥ 7%), nearly all patients showed clinically important increases in HbA1c (91.9%). Conversely, among patients who improved (from HbA1c ≥ 7% to < 7%), 99.1% achieved a clinically important reduction (Δ > 0.5) (Table 3 and Fig 2).

The left panel shows categorical changes in HbA1c relative to the 7% threshold between baseline and follow-up: worsened (<7% to ≥7%), maintained ≥7%, maintained <7%, and improved (≥7% to <7%). The right panel displays the

**Table 1. Baseline characteristics of the source population and included sample.**

| | Sampling frame | Analyzed sample | p-value | % missing |
|---|---|---|---|---|
| DEMOGRAPHICS | | | | |
| Age ≥ 60 years | 3384/ 6577 (51.5) | 449/ 741 (60.6) | **0.001** | 0% |
| Male sex | 2348/ 6577 (35.7) | 225/ 741 (30.4) | **0.005** | 0% |
| Primary education or less | 2675/ 6033 (55.7) | 350/ 672 (52.1) | **0.001** | 9.4% |
| Medical history | | | | |
| Diabetes duration >10 years | 1675/ 6577 (25.5) | 212/ 741 (28.6) | 0.070 | 0% |
| Hypertension | 1499/ 6577 (22.8) | 182/ 741 (24.6) | 0.298 | 0% |
| Dyslipidemia | 842/ 6577 (12.8) | 136/ 741 (18.4) | **0.001** | 0% |
| Tuberculosis | 261/ 6577 (4.0) | 25/ 741 (3.4) | 0.489 | 0% |
| Previous diabetes-related hospitalization | 217/ 6577 (4.3) | 17/ 741 (2.3) | 0.172 | 0% |
| Peripheral neuropathy | 386/ 6577 (5.9) | 50/ 741 (6.8) | 0.381 | 0% |
| Non-proliferative retinopathy | 147/ 6577 (2.2) | 21/ 741 (2.8) | 0.366 | 0% |
| Proliferative retinopathy | 45/ 6577 (0.7) | 8/ 741 (1.1) | 0.329 | 0% |
| Diabetic foot without amputation | 132/ 6577 (2.0) | 17/ 741 (2.3) | 0.698 | 0% |
| Diabetic foot with amputation | 118/ 6576 (1.8) | 12/ 741 (1.6) | 0.845 | 0% |
| Ischemic heart disease | 35/ 6577 (0.5) | 8/ 741 (1.1) | 0.111 | 0% |
| Stroke | 74/ 6577 (1.1) | 10/ 741 (1.4) | 0.718 | 0% |
| Peripheral artery disease | 28/ 6577 (0.4) | 6/ 741 (0.8) | 0.241 | 0% |
| Hypoglycemia | 35/ 6542 (0.5) | 5/ 739 (0.7) | 0.817 | |
| CLINICAL EVALUATION | | | | |
| BMI ≥ 30 kg/m² | 2130/ 6404 (33.3) | 234/ 733 (31.9) | 0.492 | 1.1% |
| Abdominal circumference ≥102 cm (men) or ≥88 cm (women) | 1685/ 2897 (58.2) | 214/ 368 (58.1) | 1.000 | 50.4% |
| Diabetes medication | | | | |
| Individual drugs | | | | |
| Metformin | 4343/6577 (66.0) | 526/ 741 (70.9) | **0.008** | 0% |
| Sulfonylureas | 237/ 6577 (3.6) | 25/ 741 (3.4) | 0.830 | 0% |
| DPP4 inhibitors | 30/ 6577 (0.5) | 4/ 741 (0.5) | 0.974 | 0% |
| Human insulin | 1272/ 6577 (19.3) | 146/ 741 (19.7) | 0.851 | 0% |
| Insulin analogues | 479/ 6577 (7.3) | 63/ 741 (8.5) | 0.259 | 0% |
| Any insulin therapy | 1732/ 6577 (26.3) | 208/ 741 (28.1) | 0.331 | 0% |
| Glitazones | 15/ 6577 (0.2) | 3/ 741 (0.4) | 0.596 | 0% |
| SGLT2 inhibitors | 14/ 6577 (0.2) | 3/ 741 (0.4) | 0.556 | 0% |
| GLP-1 agonists | 0/ 6577 (0.03) | 0/ 741 (0.0) | 1.000 | 0% |
| Initial treatment regimen | | | | |
| No medication | 1006/ 6577 (15.3) | 83/ 741 (11.2) | **0.004** | 0% |
| OADs | 3839/ 6577 (58.4) | 450/ 741 (60.7) | 0.231 | 0% |
| Insulin only | 1157/ 6577 (17.6) | 127/ 741 (17.1) | 0.798 | 0% |
| Insulin + other drugs | 575/ 6577 (8.7) | 81/741 (10.9) | 0.056 | |
| Laboratory parameters | | | | |
| Fasting glucose ≥130 mg/dL | 3637/ 5834 (62.3) | 413/ 688 (60.0) | 0.253 | 7.2% |
| 2-hour postprandial glucose ≥180 mg/dL | 582/ 1198 (48.6) | 64/154 (41.6) | 0.119 | 79.3% |
| HbA1c ≥ 7% | 1917/ 2931 (65.4) | 494/ 741 (66.7) | 0.546 | 0% |
| Total cholesterol ≥200 mg/dL | 1091/ 2771 (39.4) | 220/ 551 (39.9) | 0.844 | 25.9% |
| HDL < 40 mg/dL (men) or <50 mg/dL (women) | 1516/ 2547 (59.5) | 315/ 527 (59.8) | 0.953 | 28.9% |
| LDL ≥ 100 mg/dL | 1917/ 2646 (72.5) | 389/ 546 (71.3) | 0.604 | 26.4% |

*(Continued)*

**Table 1.** (Continued)

| | Sampling frame | Analyzed sample | p-value | % missing |
|---|---|---|---|---|
| Triglycerides ≥150 mg/dL | 1273/ 2578 (49.4) | 254/ 500 (50.8) | 0.594 | 32.6% |
| Albuminuria ≥30 mg/24 h | 285/ 1003 (28.4) | 51/ 208 (24.5) | 0.291 | 71.9% |
| Creatinine ≥1 mg/dL | 427/ 2480 (17.2) | 80/ 483 (16.6) | 0.777 | 34.9% |
| Creatinine clearance <60 mL/min | 296/2477 (11.9) | 54/482 (14.2) | 0.699 | 34.9% |

Values are presented as n/N (%). Comparisons between the source population and the included sample were performed using the χ² test or Fisher's exact test, as appropriate. The p-value indicates the difference between the source population and the included sample. % missing refers to the proportion of records with missing information for each variable relative to the total. BMI: body mass index; OADs: Oral antidiabetic drugs only

**Table 2.** Changes in glycemic control (HbA1c) from baseline to follow-up: proportions, absolute differences, and paired comparisons.

| | Outcome | p-value |
|---|---|---|
| Proportion of patients with poor glycemic control (HbA1c ≥ 7%) | | |
| Proportion of poor glycemic control | | |
| Baseline HbA1c ≥ 7%, n/N (%) | 494/741 (66.6) | |
| Final HbA1c ≥ 7%, n/N (%) | 450/741 (60.7) | |
| Absolute difference (baseline–follow-up), % (95% CI) | 5.9 (2.4–9.4) [a] | |
| Paired analysis by HbA1c category change | | **<0.001** [b] |
| Maintained HbA1c ≥ 7%, n/N (%) | 388/741 (52.4) | |
| Maintained HbA1c < 7%, n/N (%) | 185/741 (25.0) | |
| Improved (≥7% a <7%), n/N (%) | 106/741 (14.3) | |
| Worsened (<7% a ≥7%), n/N (%) | 62/741 (8.4) | |
| Paired OR (IC 95%) | 1.71 (1.23–2.37) [c] | |
| Paired analysis by numerical change in HbA1c | | |
| Paired assessment according to the direction of the baseline-final difference | | **0.002** [d] |
| Positive difference (HbA1c decreased), n/N (%) | 322/741 (43.4) | |
| No change, n/N (%) | 153/741 (20.6) | |
| Negative difference (HbA1c increased), n/N (%) | 266/741 (35.9) | |
| Individual HbA1c change, (%) | | |
| Baseline HbA1c, median [IQR] | 8.3 [6.3 to 10.9] | |
| Follow-up HbA1c, median [IQR] | 8.1 [6.2 to 10.9] | |
| Numeric | | |
| Absolute individual change, median [IC95%] | 0 (−0.03 to 0.03) [e] | 1.000 [f] |
| Categorized by clinical relevance of change | | |
| Δ > 0.5g, n/N (%) | 264/741 (35.6) | |
| Δ −0.5 to 0.5 n/N (%) | 267/741 (36.0) | |
| Δ < −0.5, n/N (%) | 210/741 (28.3) | |

[a] Absolute difference in proportions with 95% CI; positive values indicate a reduction in the proportion of patients with HbA1c ≥ 7% at follow-up. CI calculated considering paired design. [b] McNemar test, $H_0$: difference in discordant pairs = 0. [c] Paired OR calculated as the ratio of patients in discordant cells (improved/worsened). OR >1 indicates exposure (HbA1c ≥ 7%) more frequent at baseline; OR <1 indicates more frequent at follow-up. [d] Paired Wilcoxon signed-rank test; $H_0$: median paired difference = 0. [e] Median and 95% CI of absolute baseline–follow-up HbA1c change estimated by quantile regression with robust variance. [f] Wald test, $H_0$: median = 0. [g] Clinically relevant HbA1c changes expected following interventions in hospital diabetes programs.

**Table 3. Relationship between categorical change in glycemic control and the magnitude of individual HbA1c change (baseline to follow-up).**

|  | Maintained <7% | Maintained ≥7% | Worsened <7% to ≥7% | Improved ≥7% to <7% |
|---|---|---|---|---|
| Δ<−0.5 | 30 (16.2) | 123 (31.7) | 57 (91.9) | 0 |
| Δ −0.5 to 0.5 | 121 (65.4) | 140 (36.1) | 5 (8.1) | 1 (0.9) |
| Δ>0.5 | 34 (18.4) | 125 (32.2) | 0 | 105 (99.1) |
| Total | 185 (100) | 388 (100) | 62 (100) | 106 (100) |

Δ corresponds to the difference in HbA1c (follow-up minus baseline). "Improved" indicates a change from HbA1c≥7% to <7%; "Worsened" indicates a change from <7% to ≥7%. "Maintained <7%" represents patients with both measurements <7%, and "Maintained ≥7%" those with both ≥7%. The magnitude of individual change was classified into three categories using ±0.5% cutoffs: clinically relevant increase (Δ<−0.5%), minimal/no change (Δ −0.5 to 0.5%), and clinically relevant decrease (Δ>0.5%). Data are presented as n (%).

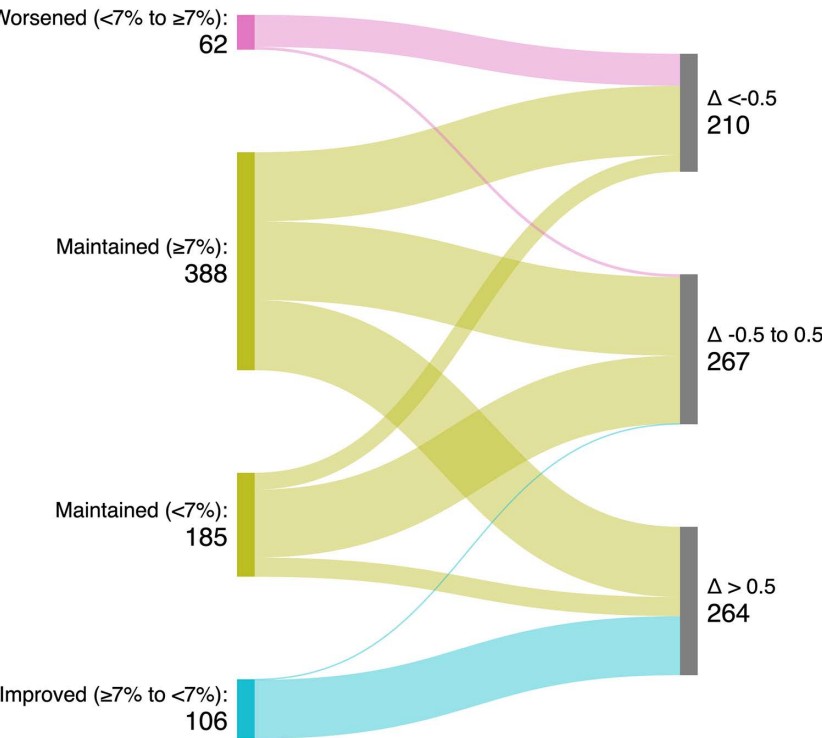

**Fig 2. Changes in glycemic control: HbA1c-based categories and individual absolute variation.**

individual absolute change in HbA1c (Δ), classified as: clinically relevant increase (Δ<−0.5%), minimal/no change (Δ −0.5% to 0.5%), and clinically relevant decrease (Δ>0.5%).

Supplementary analyses further explored the dynamics of HbA1c changes. **S1 Table** and **S1 Fig** display the distribution of absolute differences in HbA1c between baseline and follow-up. Most patients clustered around values close to zero, with 36% showing no clinically important change (−0.5 to +0.5%), whereas 28.3% experienced reductions ≥0.5% and 35.6% exhibited increases ≥0.5%.

S2 Table, S3 and S4 Figs illustrate transitions across HbA1c categories. Although a substantial proportion of patients remained in poor glycemic control (≥7%), meaningful flows toward improvement (≥7% to <7%) were observed, reinforcing the consistency between categorical shifts and the absolute magnitude of HbA1c variation.

## Factors associated with poor glycemic control in GEE models

Longitudinal analyses using GEE models showed a statistically significant reduction in the proportion of patients with HbA1c ≥ 7% at one-year follow-up (adjusted OR: 0.73; 95% CI: 0.61–0.88; p = 0.001). This finding indicates that, on average, the odds of poor glycemic control at follow-up were 27% lower than at baseline, independently of the other covariates included in the model. Regarding associated factors, patients with a diabetes duration of ≥10 years had an adjusted OR of 2.58 (95% CI: 1.82–3.64; p < 0.001), indicating more than a twofold higher likelihood of maintaining HbA1c ≥ 7% compared with those with a shorter disease duration. In terms of treatment regimen, insulin use, either as monotherapy (adjusted OR: 5.99; 95% CI: 3.65–9.82; p < 0.001) or in combination with oral antidiabetic agents (adjusted OR: 5.19; 95% CI: 3.08–8.77; p < 0.001), was strongly associated with an increased probability of poor glycemic control.

Regarding metabolic factors, hypertriglyceridemia was associated with a higher likelihood of poor glycemic control (HbA1c ≥ 7%) (adjusted OR: 2.06; 95% CI: 1.46–2.89; p < 0.001), a pattern that was also observed among participants with missing data for this variable. Finally, patients with an estimated creatinine clearance <60 mL/min showed a lower probability of persistent poor glycemic control (adjusted OR: 0.54; 95% CI: 0.30–0.96; p = 0.038). In contrast, variables such as age, sex, educational level, hypertension, and BMI-defined obesity were not significantly associated with poor glycemic control after multivariable adjustment (Table 4).

The supplementary GEE regression models confirmed the robustness of the main findings. Insulin use (either alone or in combination with oral agents), diabetes duration ≥10 years, and hypertriglyceridemia remained consistent predictors of poor glycemic control (HbA1c ≥ 7%), regardless of model specification and the approach used to handle missing data (S3 Table).

Similarly, in the sensitivity analysis restricted to complete cases (i.e., without creating a "missing data" category), the time effect remained statistically significant in Models 1 and 2, indicating a lower probability of HbA1c ≥ 7% at follow-up compared with baseline. In Model 3 (n = 312), the time effect lost statistical significance, which was expected given the substantial reduction in sample size; however, both direction and magnitude of the estimate were consistent with those observed in the previous models, supporting the overall stability of the main finding (S4 Table).

## Discordant-pair analysis: Factors associated with improvement

In the analysis restricted to discordant pairs using the McNemar test, a significant reduction in poor glycemic control at one year was confirmed (OR = 1.71; 95% CI: 1.23–2.37; p < 0.001), indicating that transitions toward improvement (HbA1c ≥ 7% to <7%) were more frequent than transitions toward deterioration (<7% to ≥7%). Stratified analyses revealed stronger associations in specific subgroups. Improvement was more pronounced among patients aged ≥60 years (OR = 1.81; 95% CI: 1.18–2.79; p = 0.005) and among men (OR = 2.18; 95% CI: 1.17–4.23; p = 0.011). Similarly, patients with a diabetes duration of less than 10 years were more likely to improve glycemic control (OR = 1.76; 95% CI: 1.22–2.58; p = 0.002). Regarding clinical characteristics, individuals with a body mass index <30 kg/m2 showed a higher likelihood of improvement (OR=2.24; 95% CI: 1.42–3.60; p = 0.001). At the laboratory level, the absence of microalbuminuria was associated with greater odds of improvement (OR=1.71; 95% CI: 1.13–2.62; p = 0.010), as was preserved renal function, defined as creatinine clearance ≥60 ml/min (OR=3.00; 95% CI: 1.37–7.25; p = 0.003) (Table 5).

## Supplementary analyses

In additional analyses, the distribution of individual HbA1c changes across demographic and clinical subgroups (S5 Table) showed no meaningful differences, reinforcing the interpretation that the overall changes observed reflect cohort-level evolution rather than effects driven by specific subpopulations.

Consistently, S6 and S7 Tables, where HbA1c change was categorized using clinically important thresholds (≥0.5%), confirmed that the observed trends were largely homogeneous across strata. Statistically significant differences emerged

**Table 4. Demographic, clinical, therapeutic, and laboratory factors associated with changes in the proportion of patients with HbA1c ≥ 7% between baseline and follow-up: crude and adjusted GEE regression analyses.**

| | Absolute difference Baseline–Follow-up hbA1c ≥ 7% Δ (95% CI) | Crude OR (95% CI) | p-value | Adjusted OR (95% CI) | p-value |
|---|---|---|---|---|---|
| Overall (time effect) | | **0.88 (0.78 −0.98)** | **0.022** | **0.73 (0.61 −0.88)** | **0.001** |
| **Age** | | | | | |
| <60 years | 5.1% (−0.7 to 10.9) | 1.00 | | 1.00 | |
| ≥60 years | 6.5% (1.9 to 11.0) | 0.85 (0.66 −1.11) | 0.241 | 0.77 (0.57–1.05) | 0.104 |
| **Sex** | | | | | |
| Female | 4.8% (0.5 to 9.1) | 1.00 | | 1.00 | |
| Male | 8.4% (1.8 to 15.0) | 1.09 (0.82 −1.45) | 0.543 | 1.01 (0.73–1.39) | 0.944 |
| **Education** | | | | | |
| Primary or less | 6.5% (1.0 to 12.1) | 1.00 | | 1.00 | |
| Secondary or higher | 4.3 (−0.8 to 9.5) | 1.12 (0.85–1.48) | 0.392 | 1.12 (0.31–1.53) | 0.439 |
| Unknown (9.4%) | 10.1 (−1.2 to 21.5) | 1.17 (0.72–1.91) | 0.500 | 1.41 (0.88–2.29) | 0.153 |
| **Diabetes duration** | | | | | |
| <10 years | 6.8% (2.4 to 11.1) | 1.00 | | 1.00 | |
| ≥10 years | 3.7% (−0.2 to 9.9) | **2.72 (1.98–3.75)** | **<0.001** | **2.58 (1.82–3.64)** | **<0.001** |
| **Hypertension** | | | | | |
| No | 7.1% (3.1 to 11.2) | 1.00 | | 1.00 | |
| Yes | 2.1% (−5.4 to 9.8) | 0.85 (0.63–1.15) | 0.313 | 1.07 (0.77–1.48) | 0.689 |
| **Tuberculosis** | | | | | |
| No | 6.2% (2.7 to 9.9) | 1.00 | | | |
| Yes | −0.045 (−21.4 to 13.4) | 1.64 (0.71–3.78) | 0.240 | | |
| **Abdominal obesity** | | | | | |
| No | 7.8% (0.7 to 14.8) | 1.00 | | | |
| Yes | 3.7% (−3.4 to 10.9) | 0.95 (0.65–1.39) | 0.821 | | |
| Missing data(50.4%) | 6.4% (−13.2 to 0.4) | − | | | |
| **Obesity** | | | | | |
| BMI < 30 kg/m2 | 7.2% (3.2 to 11.1) | 1.00 | | 1.00 | |
| BMI ≥ 30 kg/m2 | 3.4% (−4.0 to 10.8) | 0.72 (0.55–0.94) | 0.020 | 0.91 (0.67–1.23) | 0.531 |
| Unknown (1.1%) | 0% (−61.5 to 61.4) | 0.85 (0.31–2.30) | 0.754 | 1.09 (0.38–3.08) | 0.689 |
| **Treatment regimen** | | | | | |
| None | 10.8 (−1.4 to 23.1) | 0.96 (0.64–1.43) | 0.853 | 0.94 (0.61–1.45) | 0.794 |
| OADs | 5.5 (0.7 to 10.4) | 1.00 | | 1.00 | |
| Insulin only | 3.9 (−2.7 to 10.6) | **5.99 (3-70–9.69)** | **<0.001** | **5.99 (3.65–9.82)** | **<0.001** |
| Insulin plus OADs | 6.1 (−4.3 to 16.7) | **5.59 (3.31–9.43)** | **<0.001** | **5.19 (3.08–8.77)** | **<0.001** |
| **Hypertriglyceridemia** | | | | | |
| No | 6.5% (≈0.0 to 13.1) | 1.00 | | 1.00 | |
| Yes | 5.1% (−0.9 to 11.0) | **1.75 (1.27–2.40)** | **0.001** | **2.06 (1.46–2.89)** | **<0.001** |
| Missing data (32.6%) | 6.2 (≈0.0 to 12.5) | 1.42 (1.03–1.95) | 0.028 | 1.58 (1.11–2.23) | 0.010 |
| **eGFR < 60 mL/min/1.73 m2** | | | | | |
| No | 6.3% (1.5 to 11.2) | 1.00 | | 1.00 | |
| Yes | 0% (−16.4 to 16.4) | 0.74 (0.46–1.21) | 0.233 | **0.54 (0.30–0.96)** | **0.038** |
| Missing data (34.9%) | 6.5 (−14.8 to 1.7) | 0.97 (0.73–1.30) | 0.888 | 0.87 (0.64–1.18) | 0.364 |
| **Microalbuminuria** | | | | | |
| No | 11.5% (3.6 to 19.4) | 1.00 | | | |

*(Continued)*

**Table 4.** (Continued)

| | Absolute difference Baseline–Follow-up hbA1c ≥ 7% Δ (95% CI) | Crude OR (95% CI) | p-value | Adjusted OR (95% CI) | p-value |
|---|---|---|---|---|---|
| Yes | 11.8% (−3.1 to 26.6) | 0.77 (0.43–1.35) | 0.364 | | |
| Missing data (71.9%) | 3.7 (≈0.0 to 7.9) | – | – | | |

Abbreviations: GEE, Generalized Estimating Equations; OR, odds ratio; CI, confidence interval; eGFR, estimated glomerular filtration rate. BMI: body mass index; OAD: Oral antidiabetic drugs only; eGFR:Estimated Glomerular filtration rate. Absolute baseline–follow-up differences are expressed as percentages with 95% CI. ORs were estimated using binomial GEE models with logit link, exchangeable correlation structure, and robust standard errors. Adjusted models included demographic, clinical, therapeutic, and relevant metabolic variables.

only when stratified by baseline glycemic status (HbA1c < 7% vs. ≥ 7%), highlighting baseline control as the primary determinant of subsequent change.

## Discussion

### Main findings

In this study, we observed a reduction in the proportion of patients with poor glycemic control over the follow-up period. Longitudinal analyses using GEE models showed that the probability of having HbA1c ≥ 7% at one year was approximately one-third lower than at baseline. Consistently, paired analyses indicated that transitions from HbA1c ≥ 7% to < 7% were more frequent than transitions in the opposite direction.

Despite this overall improvement, more than half of the cohort remained with HbA1c ≥ 7% at the end of follow-up, indicating that the magnitude of improvement was modest. Longer duration of diabetes, insulin use, and hypertriglyceridemia were independently associated with a higher likelihood of persistent poor glycemic control, whereas an estimated glomerular filtration rate <60 mL/min was associated with a lower probability of HbA1c ≥ 7%.

### Comparison with previous studies

Our findings are consistent with previous studies conducted in Peru, Latin America, and other regions. In the Peruvian context, hospital-based studies have consistently reported a low proportion of patients achieving HbA1c < 7%, with estimates of 31.7% in public hospitals [12], 25.3% in EsSalud [13] and 35.8% at the national level [5]. These figures are comparable to the baseline levels of glycemic control observed in our cohort.

Notably, López-Huamanrayme et al. reported that 10.2% of patients improved their glycemic control, while 5.1% experienced deterioration after one year of follow-up [9]. This pattern closely mirrors both direction and magnitude of change observed in our study, supporting the consistency of these findings across different Peruvian clinical settings.

In Latin America, a similar pattern has been reported. In Mexico, only 39–43% of patients achieved an HbA1c < 7%, with a progressive decline in glycemic control as diabetes duration increases [14]. Likewise, a multicenter study conducted across nine countries found that only 43% of patients met glycemic targets, with substantially poorer control among those with more than 15 years of disease duration [15]. Although targeted interventions using advanced technologies, such as *mHealth* strategies implemented in Argentina, have demonstrated meaningful improvements, their scalability and generalizability to routine clinical practice remain uncertain [16].

Similarly, systematic reviews indicate that poor glycemic control remains the global norm, with reported prevalences ranging from 45% to as high as 93% [17,18]. Educational and dietary interventions have been associated with modest reductions in HbA1c (approximately –0.3% to –0.6%), while more structured dietary approaches achieve larger decreases (–0.7% to –1.3%). However, sustaining these improvements over time continues to represent a major challenge [19,20].

**Table 5. Demographic, clinical, therapeutic, and laboratory factors associated with changes in glycemic control among discordant pairs (HbA1c ≥ 7% to <7% transitions between baseline and follow-up).**

| | ≥7% to ≥7% | ≥7% to <7% | <7% to ≥7% | <7% to <7% | OR (95% CI) | p (Mc Nemar) |
|---|---|---|---|---|---|---|
| General | 388 (52.4) | 106 (14.3) | 62 (8.4) | 185 (24.9) | **1.71 (1.23–2.37)** | **<0.001** |
| **Age** | | | | | | |
| <60 years | 159 (54.5) | 41 (14.0) | 26 (8.9) | 66 (22.6) | 1.58 (0.94–2.69) | 0.086 |
| ≥60 years | 229 (51.0) | 65 (14.5) | 36 (8.0) | 119 (26.5) | **1.81 (1.18–2.79)** | **0.005** |
| **Sexo** | | | | | | |
| Female | 267 (51.7) | 71 (13.8) | 46 (8.9) | 132 (25.6) | **1.54 (1.05–2.28)** | **0.026** |
| Male | 121 (53.8) | 35 (15.6) | 16 (7.1) | 53 (23.6) | **2.18 (1.17- 4.23)** | **0.011** |
| **Education** | | | | | | |
| Primary or less | 173 (49.4) | 56 (16.0) | 33 (9.4) | 88 (25.1) | **1.69 (1.08–2.69)** | **0.019** |
| Secondary or higher | 176 (54.7) | 40 (12.4) | 26 (8.1) | 80 (24.8) | 1.53 (0.91–2.62) | 0.108 |
| Unknown | 39 (56.5) | 10 (14.5) | 3 (4.3) | 17 (24.6) | 3.33 (0.85–18.8) | 0.092 |
| **Diabetes duration** | | | | | | |
| <10 years | 240 (45.4) | 83 (15.7) | 47 (8.9) | 159 (30.1) | **1.76 (1.22–2.58)** | **0.002** |
| ≥10 years | 148 (69.8) | 23 (10.8) | 15 (7.1) | 26 (12.3) | 1.53 (0.76–3.15) | 0.256 |
| **Hypertension** | | | | | | |
| No | 299 (53.5) | 82 (14.7) | 42 (7.5) | 136 (24.3) | **1.95 (1.33 −2.90)** | **0.001** |
| Yes | 89 (48.9) | 24 (13.2) | 20 (11.0) | 49 (26.9) | 1.20 (0.63–2.29) | 0.652 |
| **Tuberculosis** | | | | | | |
| No | 371 (51.8) | 105 (14.7) | 60 (8.4) | 180 (25.1) | **1.75 (1.26–2.44)** | **0.001** |
| Yes | 17 (68.0) | 1 (4.0) | 2 (8.0) | 5 (20.0) | 0.5 (0.01–9.60) | 1.000 |
| **Abdominal obesity** | | | | | | |
| No | 85 (55.2) | 19 (12.3) | 7 (4.5) | 43 (27.9) | **2.71 (1.09–7.64)** | **0.029** |
| Yes | 107 (50.0) | 31 (14.5) | 23 (10.7) | 53 (24.8) | 1.34 (0.76–2.42) | 0.341 |
| Missing | 196 (52.5) | 56 (15.0) | 32 (8.6) | 89 (23.9) | **1.75 (1.11–2.79)** | **0.014** |
| **Obesity** | | | | | | |
| BMI < 30 kg/m2 | 283 (56.7) | 65 (13.0) | 29 (5.8) | 122 (24.4) | **2.24 (1.42 −3.60)** | **0.001** |
| BMI ≥ 30 kg/m2 | 102 (43.6) | 39 (16.7) | 31 (13.2) | 62 (26.5) | 1.25 (0.76–2.08) | 0.403 |
| Missing | 3 (37.5) | 2 (25.0) | 2 (25.0) | 1 (12.5) | 1.00 (0.07–13.8) | 1.000 |
| **Diabetes treatment** | | | | | | |
| None | 33 (39.8) | 16 (19.3) | 7 (8.4) | 27 (32.5) | 2.28 (0.88 −6.56) | 0.093 |
| Oral drugs only | 188 (41.8) | 70 (15.6) | 45 (10.0) | 147 (32.7) | **1.56 (1.05–2.06)** | **0.025** |
| Insulin only | 104 (81.9) | 10 (7.9) | 5 (3.9) | 8 (6.3) | 2.00 (0.62–7.45) | 0.302 |
| Insulin + oral drugs | 63 (77.8) | 10 (12.4) | 5 (6.2) | 3 (3.5) | 2.00 (0.62–7.45) | 0.302 |
| **Hypertriglyceridemia** | | | | | | |
| No | 108 (43.9) | 39 (15.9) | 23 (9.3) | 76 (30.9) | 1.70 (0.99–2.97) | 0.055 |
| Yes | 150 (59.1) | 33 (13.0) | 20 (7.9) | 51 (20.1) | 1.65 (0.92–3.03) | 0.098 |
| Missing | 130 (53.9) | 34 (14.1) | 19 (7.9) | 58 (24.1) | 1.78 (0.99–3.32) | 0.053 |
| **Microalbuminuria** | | | | | | |
| No | 84 (53.5) | 27 (17.2) | 9 (5.7) | 37 (23.6) | **1.71 (1.13–2.62)** | **0.010** |
| Yes | 24 (47.1) | 9 (17.6) | 3 (5.9) | 15 (29.4) | 1.00 (0.33–3.06) | 1.000 |
| Missing | 280 (52.5) | 70 (13.1) | 50 (9.4) | 133 (25.0) | | 1.000 |
| **eGFR < 60 mL/min/1.73 m2** | | | | | | |
| No | 224 (52.3) | 65 (15.2) | 38 (8.9) | 101 (23.6) | **3.00 (1.37–7.25)** | **0.003** |
| Yes | 23 (42.6) | 8 (14.8) | 8 (14.8) | 15 (27.8) | 3.00 (0.75–17.2) | 0.146 |
| Missing | 224 (52.3) | 65 (15.2) | 38 (8.9) | 101 (23.6) | 1.40 (0.96–2.05) | 0.082 |

BMI: body mass index; OAD: Oral antidiabetic drugs only; eGFR:Estimated Glomerular filtration rate. Analysis restricted to discordant pairs, i.e., patients who changed HbA1c category between baseline and follow-up (≥7% ↔ <7%). McNemar test was applied within each category of explanatory variables. OR (odds ratio) with 95% CI and p-values are reported.

Finally, population-based cohorts reveal heterogeneous glycemic trajectories over time. In the United States, most patients (95.5%) maintain stable glycemic control over a two-year period, although a small but relevant proportion (4.4%) experiences rapid deterioration [21]. In Japan, mean population HbA1c levels remained largely stable between 2012 and 2019; however, poorer glycemic control has been observed among younger, insulin-treated individuals [22]. In contrast, data from the SEARCH study indicate a worsening trend in glycemic control over the past decade among adolescents and young adults [23].

Taken together, these findings place our study within the pattern consistently reported in both national and international literature: modest improvements in average glycemic control coexisting with a substantial proportion of patients who remain above recommended therapeutic targets. The magnitude of change observed, as well as the identification of subgroups with poorer trajectories, particularly those with longer disease duration and insulin use, aligns with evidence from hospital-based studies and population cohorts. Collectively, these results reinforce the notion that, in the absence of intensive interventions or broader access to modern therapies, improvements in glycemic control tend to be modest and unevenly distributed across patient populations.

## Explanation of findings and biological plausibility

The observed reduction in the proportion of patients with HbA1c ≥ 7% over the follow-up period suggests an overall improvement in glycemic control at the population level [24]. This finding is consistent with the notion that regular contact with a specialized care service facilitates progressive therapeutic adjustments, pharmacological intensification, and reinforcement of non-pharmacological recommendations [25]. However, this average effect does not imply a uniform improvement across individuals. This heterogeneity is reflected in the null median change in individual HbA1c values and in the coexistence of distinct trajectories, with subgroups of patients who improved, worsened, or remained stable over time [26].

In our analysis, the magnitude of the observed effect varied according to the statistical approach used. The paired McNemar odds ratio indicated a moderate effect in favor of improvement (OR=1.71), whereas the adjusted GEE model revealed a more robust and time-consistent effect (OR=0.73). It is important to note that, within the GEE framework, odds ratios below 1 should be interpreted as protective against poor glycemic control, as this method estimates population-averaged odds ratios while accounting for within-subject correlation over time. The apparent discrepancy between both approaches can be explained by their underlying assumptions: the McNemar test is restricted to discordant pairs only, whereas GEE models incorporate the full longitudinal structure of the data and allow for multivariable adjustment, resulting in more stable and generalizable estimates (29). These findings further support the interpretation that glycemic improvement occurs at the population level, albeit with substantial interindividual heterogeneity.

Among the factors associated with poor glycemic control, a diabetes duration of ≥10 years (adjusted OR=2.58) is consistent with the progressive decline in pancreatic β-cell function, which limits the ability of conventional therapeutic regimens to sustain glycemic targets over the long term. The use of insulin, either alone or in combination with oral antidiabetic agents (adjusted ORs 5–6), should be interpreted primarily as a marker of greater disease severity and prior failure of oral therapy, rather than as an intrinsic adverse effect of insulin itself. Likewise, hypertriglyceridemia (adjusted OR 2.06) is biologically plausible, as it reflects an underlying milieu of insulin resistance, lipotoxicity, and coexistence of other components of the metabolic syndrome, conditions that collectively hinder the achievement and maintenance of adequate glycemic control.

Conversely, the lower probability of poor glycemic control observed among patients with reduced glomerular filtration may be explained by several non-mutually exclusive mechanisms. These include the use of more conservative pharmacological regimens, closer clinical follow-up, more frequent dose adjustments, and, in some cases, reduced caloric intake associated with chronic kidney disease. In addition, in more advanced stages of renal impairment, decreased insulin clearance prolongs insulin action, thereby facilitating lower plasma glucose levels. Finally, conditions commonly accompanying chronic kidney disease—such as anemia and reduced erythrocyte lifespan may influence HbA1c values [27],

leading to lower measured levels that do not necessarily reflect true improvements in glycemic control. Nevertheless, this finding should be interpreted with caution, given the high proportion of missing data for this variable (34.9%), which may have influenced the final estimates.

Finally, the fact that our data were derived from a public tertiary referral hospital places these results within a setting characterized by greater clinical complexity [28]. Such environments tend to concentrate patients with longer disease duration, more frequent insulin use, and multiple comorbidities [29]. This higher burden of complex cases may partly explain the differences observed when compared with population-based cohorts and may have attenuated the magnitude of the improvements in glycemic control identified in this study [30].

## Public health relevance and practical implications

During the study period, the therapeutic options available within the Ministry of Health (MINSA) system were limited to metformin, sulfonylureas, and NPH and regular insulin. Within this constrained therapeutic context, it is noteworthy that although a statistically significant reduction in glycated hemoglobin category transitions was observed, a substantial proportion of patients failed to achieve the HbA1c target of <7%. This indicates that, despite a favorable overall trend, gains in optimal glycemic control were modest. Given that the prevention of chronic microvascular complications; such as diabetic retinopathy, nephropathy, and neuropathy; requires sustained glycemic control over time [31], the one-year outcomes observed in this study offer a limited outlook regarding the long-term health impact for these patients.

Importantly, glycemic control, while essential, represents only one component of comprehensive metabolic management. International evidence consistently shows that the simultaneous control of glucose, blood pressure, and lipid levels is what ultimately determines the risk of macrovascular complications and mortality [32] If achieving adequate HbA1c targets alone is already challenging, attaining concurrent control of all three risk factors is even more complex. From a public health perspective, these findings underscore the need to strengthen access to therapies with improved efficacy and safety profiles, promote multidisciplinary care models, and implement intensive follow-up programs that integrate all components of metabolic risk management, particularly in resource-limited health systems.

## Strengths and limitations

This study has several limitations that should be considered when interpreting its findings. First, although the database initially included 6,577 individuals with type 2 diabetes mellitus, only 741 met the eligibility criteria and had sufficient information for the main analysis. While this represents just over 10% of the original sampling frame, the clinical registry itself holds substantial potential for future research, provided that the completeness and quality of longitudinal data are further improved. The relatively limited sample size may have reduced the ability to detect associations of smaller magnitude and should be taken into account when interpreting the results. In addition, comparisons between the included sample and the overall sampling frame revealed differences in certain demographic and clinical characteristics, such as a higher proportion of individuals aged ≥60 years and differences in metformin use, which may have influenced the estimated associations. These discrepancies raise the possibility of selection bias and warrant caution when extrapolating the findings to the entire population receiving care in the endocrinology service.

Second, several variables had incomplete data. As a predefined strategy, variables with more than 35% missing values were excluded from the multivariable models, as was the case for abdominal obesity (50.4%) and microalbuminuria. However, for the anthropometric domain, body mass index was available and used as an alternative measure. In addition, albuminuria mainly reflects a consequence of poor glycemic control rather than a direct causal determinant; therefore, its exclusion is unlikely to have substantially affected the main associations observed. Third, it was not possible to assess adherence to pharmacological treatment, nor to verify whether patients maintained regular attendance at endocrinology visits or accessed complementary services such as nutritional counseling or diabetes education programs. In addition, information on treatment modifications during the follow-up period was not consistently available in the clinical records,

preventing evaluation of changes in therapeutic regimens over time. These factors are likely to influence glycemic control, and their omission may have led to an underestimation of interindividual variability in treatment response. Furthermore, although the GEE models used are appropriate for longitudinal data and account for within-subject correlation, their estimates represent population-averaged effects, which limits direct inference at the individual level. Likewise, because a logit link was applied, the resulting odds ratios should be interpreted as approximations of relative risk rather than direct risk estimates. This limitation was partially addressed through complementary individual-level analyses, including paired comparisons and analyses of changes in mean HbA1c values, the results of which are presented in the supplementary tables.

In contrast, this study has several notable strengths. First, a systematic comparison between the sampling frame and the analyzed sample was conducted, allowing potential selection differences to be identified and their impact transparently addressed. Second, while a pragmatic strategy of creating a "missing data" category was used to retain sample size for variables with up to 35% missing values, complementary analyses were also performed using complete-case models, as well as models including variables with less than 10% and up to 35% missing information. A sensitivity analysis restricted exclusively to complete cases was also conducted. These approaches yielded consistent estimates, supporting the robustness of the findings. Third, the study's statistical power was evaluated across different scenarios of discordant proportions and sample sizes, confirming sufficient capacity to detect clinically relevant differences. Finally, glycemic control was assessed from multiple perspectives, including the proportion of patients with HbA1c ≥ 7%, categorical transitions, and individual HbA1c changes, providing a more comprehensive and robust understanding of glycemic patterns in this patient population.

### Research recommendations

The findings of this study highlight the need for further research on glycemic control in individuals with type 2 diabetes receiving care in public hospitals in Peru. First, future studies should incorporate longer-term follow-up to determine whether the modest reductions in HbA1c observed are sustained over time and whether they translate into meaningful reductions in micro- and macrovascular complications. Second, it is essential to include variables related to treatment adherence and continuity of care, including follow-up in endocrinology, nutrition, and diabetes education services, as these factors directly influence therapeutic effectiveness and were not assessed in the present study. Third, future research should explore the impact of newer therapeutic regimens incorporating medications with improved safety and efficacy profiles, such as GLP-1 receptor agonists or SGLT2 inhibitors, which remain limited in the public system but could substantially improve outcomes in comprehensive metabolic control. Finally, studies evaluating all three key components of metabolic management (glucose, blood pressure, and lipids) are warranted, given that controlling only one component in isolation may be insufficient to prevent major cardiovascular events.

### Conclusion

In patients with type 2 diabetes receiving care at a public referral hospital, the proportion of individuals with poor glycemic control decreased after one year of follow-up. However, the magnitude of this reduction, an absolute difference of approximately 6%, is modest from a clinical perspective. The median individual change in HbA1c was essentially zero, suggesting that the observed population-level improvement was driven primarily by a subset of patients who achieved values below the 7% threshold, while a substantial proportion experienced no clinically meaningful change. Persistence of poor glycemic control was more likely among patients with longer disease duration, insulin-based treatment regimens, and hypertriglyceridemia, whereas reduced kidney function was associated with a lower probability of HbA1c ≥ 7%, likely reflecting clinical and physiological factors related to advanced disease and treatment patterns. Overall, these findings suggest limited progress in glycemic control within the public healthcare setting and highlight the need to strengthen more intensive, individualized, and multidisciplinary approaches, as well as to expand access to effective pharmacologic options, to achieve sustained improvements in metabolic control and reduce the risk of diabetes-related complications

## Supporting information

**S1 Fig. Statistical power curves for the McNemar test.**
(TIF)

**S2 Fig. Distribution of HbA1c difference (baseline – final).**
(TIF)

**S3 Fig. Transition of HbA1c categories between baseline and final evaluation.**
(DOCX)

**S4 Fig. Baseline HbA1c and absolute change at one year.**
(DOCX)

**S1 Table. Categories of HbA1c difference (ΔHbA1c) between baseline and final.**
(DOCX)

**S2 Table. Transition matrix of HbA1c categories (baseline vs final).**
(DOCX)

**S3 Table. Factors associated with poor glycemic control (HbA1c ≥ 7%) in patients with type 2 diabetes according to different GEE model specifications.**
(DOCX)

**S4 Table. Sensitivity analysis using GEE models restricted to complete cases for the assessment of factors associated with poor glycemic control (HbA1c ≥ 7%) between baseline and final evaluation.**
(DOCX)

**S5 Table. Median individual change in HbA1c from baseline to final evaluation and differences in medians across demographic, clinical, treatment, and laboratory subgroups.**
(DOCX)

**S6 Table. Individual change in HbA1c from baseline to final evaluation categorized in the overall population according to demographic, clinical, therapeutic, and laboratory factors.**
(DOCX)

**S7 Table. Proportion of patients with a decrease or increase ≥0.5% in HbA1c according to demographic, clinical, and treatment characteristics, stratified by baseline HbA1c (<7% and ≥7%).**
(DOCX)

**S1 Dataset. Dataset A1c change.**
(XLSX)

## Author contributions

**Conceptualization:** Romina Belen Carpio-Martinez, Jose Manuel Quiroz-Eugenio, Frank Espinoza-Morales, Marlon Yovera-Aldana.

**Data curation:** Marlon Yovera-Aldana.

**Formal analysis:** Marlon Yovera-Aldana.

**Investigation:** Romina Belen Carpio-Martinez, Jose Manuel Quiroz-Eugenio, Frank Espinoza-Morales, Wendy M. Yánac-Tellería, Marlon Yovera-Aldana.

**Methodology:** Marlon Yovera-Aldana.

**Supervision:** Wendy M. Yánac-Tellería, Marlon Yovera-Aldana.

**Validation:** Marlon Yovera-Aldana.

**Writing – original draft:** Romina Belen Carpio-Martinez, Jose Manuel Quiroz-Eugenio, Frank Espinoza-Morales, Wendy M. Yánac-Tellería, Marlon Yovera-Aldana.

**Writing – review & editing:** Romina Belen Carpio-Martinez, Jose Manuel Quiroz-Eugenio, Frank Espinoza-Morales, Wendy M. Yánac-Tellería, Marlon Yovera-Aldana.

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
