## [Decision Letter · Decision Letter 0]

3 Mar 2026

Dear Dr. Yovera-Aldana,

Thank you for submitting your manuscript to PLOS ONE. After careful consideration, we feel that it has merit but does not fully meet PLOS ONE’s publication criteria as it currently stands. Therefore, we invite you to submit a revised version of the manuscript that addresses the points raised during the review process.

**ACADEMIC EDITOR:**

We have approached multiple reviewers and 2 of them have agreed and gave further comments which are reasonable to be addressed

Do consider resumitting if you are able to address them

We look forward to receiving your revised manuscript.

Kind regards,

Yee Gary Ang, MBBS MPH

Academic Editor

PLOS One

**Journal Requirements:**

https://journals.plos.org/plosone/s/file?id=wjVg/PLOSOne_formatting_sample_main_body.pdf andandandand

2. The ethical approval number(s) listed in the manuscript and/or submission metadata does not match the approval number on the ethical approval document you provided. Please ensure that all approval numbers are correct.

4. Please upload a new copy of Figure 2 and Supporting Information Figure 1 and 2  as the detail is not clear. Please follow the link for more information:  https://journals.plos.org/plosone/s/figures

5.  We note that there is identifying data in the Supporting Information file < S1 File.xlsx>. Due to the inclusion of these potentially identifying data, we have removed this file from your file inventory. Prior to sharing human research participant data, authors should consult with an ethics committee to ensure data are shared in accordance with participant consent and all applicable local laws.

-Location data

Reviewers' comments:

Reviewer's Responses to Questions

**Comments to the Author**

1. Is the manuscript technically sound, and do the data support the conclusions?

Reviewer #1: Partly

Reviewer #2: Yes

2. Has the statistical analysis been performed appropriately and rigorously?

Reviewer #1: Yes

Reviewer #2: Yes

3. Have the authors made all data underlying the findings in their manuscript fully available?

Reviewer #1: Yes

Reviewer #2: Yes

4. Is the manuscript presented in an intelligible fashion and written in standard English?

Reviewer #1: Yes

Reviewer #2: Yes

Reviewer #1: Revised according to suggestions.1. Abstract: Suggestions: Emphasize the clinical and public health relevance of the findings in the conclusion (e.g., implications for diabetes care in Peru). Clarify that the reduction was statistically significant but clinically modest, to balance interpretation.

Consider shortening sentences for readability, especially in the Methods section.

Reviewer #2: This is an important manuscript that adds to the body of information regarding level of glycaemic control and factors that influence them.

I have a few comments that will help provide more clarity to your manuscript.

Abstract: Kindly clarify at the methods section, that patients with T2DM

Results/Discussion/Limitations: Were changes in treatment assessed in this retrospective study? The authors make mention of not assessing adherence to treatment earlier however assessing changes in treatment could also help evaluate the factors associated with good or poor glycaemic control.

Others: Kindly check referencing for Line 383-384

.

Reviewer #1: No

Reviewer #2: No

---

## [Author Response · Author response to Decision Letter 1]

9 Mar 2026

Response to Journal Requirements

We thank the editorial office for the additional guidance regarding the journal requirements. We have carefully reviewed these points and revised the submission accordingly to ensure full compliance with the PLOS ONE formatting, data sharing, ethics, and figure preparation guidelines. Detailed responses to each item are provided below.

We reviewed the manuscript and ensured that it complies with the PLOS ONE formatting and style requirements, including file naming and manuscript structure according to the journal’s guidelines.

2. The ethical approval number(s) listed in the manuscript and/or submission metadata does not match the approval number on the ethical approval document you provided. Please ensure that all approval numbers are correct.

Thank you for pointing this out. The discrepancy was due to a typographical error in the manuscript, where ethical approval number was mistakenly written as 500-CIEI-CIENTÍFICA-2024 instead of 526-CIEI-CIENTÍFICA-2024. The manuscript has now been corrected to match the ethics approval document. The project (PRE-15-2024-00188) was approved by the Institutional Research Ethics Committee of Universidad Científica del Sur under Certificate No. 526-CIEI-CIENTÍFICA-2024, and also by the Institutional Ethics Committee of Hospital María Auxiliadora under Unique Registration Code HMA/CIEI/047/2024.

Thank you for pointing this out. The information entered in the Funding Information section of the submission system is correct. The discrepancy occurred in the Financial Disclosure field of the system, where the internal project code was mistakenly entered instead of the official funding resolution. At the current stage of the revision process, the Financial Disclosure field in the system is not editable by the authors. Therefore, we provide below the correct wording that should appear in the Financial Disclosure statement, consistent with the Funding Information:

Financial support for this study was partially provided by Universidad Científica del Sur (UCSUR) through the Beca Cabieses – Proyectos de Tesis de Pregrado program (grant number RD No. 004-DGIDI-CIENTIFICA-2024), awarded to the authors RBCM and JMQE. The official website of the funder is https://www.cientifica.edu.pe. The funder had no role in the study design, data collection and analysis, interpretation of results, decision to publish, or preparation of the manuscript. If the manuscript is accepted for publication, the university may provide support to cover the article processing charge (APC); this potential support will not influence any aspect of the editorial process.

We kindly ask the editorial office to update the Financial Disclosure field in the system if necessary so that it matches the Funding Information.

4. Please upload a new copy of Figure 2 and Supporting Information Figure 1 and 2 as the detail is not clear. Please follow the link for more information: https://journals.plos.org/plosone/s/figures

The requested figures (Figure 2 and Supporting Figures S1–S2) have been updated and re-uploaded with improved resolution and clarity in accordance with the PLOS ONE figure guidelines.

5. We note that there is identifying data in the Supporting Information file < S1 File.xlsx>. Due to the inclusion of these potentially identifying data, we have removed this file from your file inventory. Prior to sharing human research participant data, authors should consult with an ethics committee to ensure data are shared in accordance with participant consent and all applicable local laws. Data sharing should never compromise participant privacy. It is therefore not appropriate to publicly share personally identifiable data on human research participants. The following are examples of data that should not be shared: -Name, initials, physical address -Ages more specific than whole numbers -Internet protocol (IP) address -Specific dates (birth dates, death dates, examination dates, etc.) -Contact information such as phone number or email address -Location data -ID numbers that seem specific (long numbers, include initials, titled “Hospital ID”) rather than random (small numbers in numerical order) Data that are not directly identifying may also be inappropriate to share, as in combination they can become identifying. For example, data collected from a small group of participants, vulnerable populations, or private groups should not be shared if they involve indirect identifiers (such as sex, ethnicity, location, etc.) that may risk the identification of study participants. Additional guidance on preparing raw data for publication can be found in our Data Policy (https://journals.plos.org/plosone/s/data-availability#loc-human-research-participant-data-and-other-sensitive-data) and in the following article: http://www.bmj.com/content/340/bmj.c181.long. Please remove or anonymize all personal information (<specific identifying information in file to be removed>), ensure that the data shared are in accordance with participant consent, and re-upload a fully anonymized data set. Please note that spreadsheet columns with personal information must be removed and not hidden as all hidden columns will appear in the published file.

Thank you for this observation. We carefully reviewed the dataset to ensure compliance with PLOS data sharing policies and participant confidentiality requirements. All potential identifiers were removed from the dataset prior to resubmission. In particular, the internal record identifier column was deleted, and we verified that the dataset does not contain personal identifiers such as names, medical record numbers, contact information, dates, or location data. The fully anonymized dataset has now been re-uploaded as S1 Dataset.xlsx.

We reviewed the reviewer comments and confirmed that no specific recommendations to cite additional publications were made. No changes to the reference list were required.

Response to Reviewers

We thank the reviewers for their constructive comments and suggestions. We carefully considered all the points raised and revised the manuscript accordingly. Our responses to each comment are provided below.

Reviewer #1

1. Abstract: Suggestions: Emphasize the clinical and public health relevance of the findings in the conclusion (e.g., implications for diabetes care in Peru)

We appreciate this suggestion. The conclusion of the abstract has been revised to highlight the clinical and public health implications of our findings for diabetes management within the Peruvian public health system.

2. Clarify that the reduction was statistically significant but clinically modest, to balance interpretation.

We agree with the reviewer and have modified the conclusion of the abstract to explicitly state that although the reduction in poor glycemic control was statistically significant, the magnitude of the improvement was clinically modest.

3. Consider shortening sentences for readability, especially in the Methods section

The Methods section of the abstract has been revised to improve readability by splitting long sentences into shorter statements while preserving the methodological details.

Reviewer #2

1. Abstract: Kindly clarify at the methods section, that patients with T2DM

We thank the reviewer for this suggestion. The Methods section of the abstract has been revised to explicitly state that the study population consisted of adults diagnosed with type 2 diabetes mellitus (T2DM).

2. Were changes in treatment assessed in this retrospective study? The authors make mention of not assessing adherence to treatment earlier however assessing changes in treatment could also help evaluate the factors associated with good or poor glycaemic control

Changes in pharmacological treatment during follow-up were not systematically captured in the database used for this retrospective study. We have clarified this point in the limitations section, noting that treatment intensification or medication adjustments could influence glycemic control and should be considered in future studies.

3. Kindly check referencing for Line 383-384

Thank you for noting this. The citation was incorrectly placed on the following line. It has now been moved to appear before the final period of the sentence

---

## [Decision Letter · Decision Letter 1]

16 Mar 2026

Longitudinal changes in glycemic control and associated factors in patients with type 2 diabetes mellitus in a public referral hospital in Peru.

PONE-D-26-07014R1

Dear Dr. Yovera-Aldana,

We’re pleased to inform you that your manuscript has been judged scientifically suitable for publication and will be formally accepted for publication once it meets all outstanding technical requirements.

Kind regards,

Yee Gary Ang, MBBS MPH

Academic Editor

PLOS One

Additional Editor Comments (optional):

Reviewers' comments:

Reviewer's Responses to Questions

**Comments to the Author**

Reviewer #1: All comments have been addressed

Reviewer #2: All comments have been addressed

2. Is the manuscript technically sound, and do the data support the conclusions?

Reviewer #1: No

Reviewer #2: Yes

3. Has the statistical analysis been performed appropriately and rigorously?

Reviewer #1: Yes

Reviewer #2: Yes

4. Have the authors made all data underlying the findings in their manuscript fully available?

Reviewer #1: Yes

Reviewer #2: Yes

5. Is the manuscript presented in an intelligible fashion and written in standard English?

Reviewer #1: Yes

Reviewer #2: Yes

Reviewer #1: The research has been revised and improved to enhance its overall quality and clarity. Several sections have been carefully edited to correct previous issues and to ensure that the content is accurate and well organized. Additional information and explanations have been included to strengthen the arguments and provide better support for the research objectives. As a result of these improvements, the study is now more complete and easier for readers to understand. The structure of the research has also been refined to create a clearer flow of ideas from the introduction to the conclusion. Overall, the revised version presents the findings in a more engaging and meaningful way, making the research not only more comprehensive but also more interesting and valuable for readers and future researchers.

Reviewer #2: This manuscript adds to depth of information regarding factors affecting glycaemic control.

The authors have addressed all comments raised by the reviewer in the manuscript.

.

Reviewer #1: No

Reviewer #2: No

---

## [Editor Report · Acceptance letter]

PONE-D-26-07014R1

PLOS One

Dear Dr. Yovera-Aldana,

I'm pleased to inform you that your manuscript has been deemed suitable for publication in PLOS One. Congratulations! Your manuscript is now being handed over to our production team.

Kind regards,

on behalf of

Dr. Yee Gary Ang

Academic Editor

PLOS One